

# Recognition of inscribed cursive Pashtu numeral through optimized deep learning

Sibtain Syed[1], Khalil Khan[2], Maqbool Khan[1,3], Rehan Ullah Khan[4] and Abdulrahman Aloraini[4]

[1] Department of IT & CS, Pak-Austria Fachhochschule Institute of Applied Sciences and Technology, Haripur, KP, Pakistan
[2] Department of Computer Science, School of Engineering and Digital Sciences, Nazarbayev University, Astana, Kazakhstan
[3] PAF-IAST, Sino-Pak Center for Artificial Intelligence (SPCAI), Haripur, Pakistan
[4] Department of Information Technology, College of Computer, Qassim University, Buraydah, Saudi Arabia

## ABSTRACT

Pashtu is one of the most widely spoken languages in south-east Asia. Pashtu Numerics recognition poses challenges due to its cursive nature. Despite this, employing a machine learning-based optical character recognition (OCR) model can be an effective way to tackle this issue. The main aim of the study is to propose an optimized machine learning model which can efficiently identify Pashtu numerics from 0–9. The methodology includes data organizing into different directories each representing labels. After that, the data is preprocessed *i.e.*, images are resized to $32 \times 32$ images, then they are normalized by dividing their pixel value by 255, and the data is reshaped for model input. The dataset was split in the ratio of 80:20. After this, optimized hyperparameters were selected for LSTM and CNN models with the help of trial-and-error technique. Models were evaluated by accuracy and loss graphs, classification report, and confusion matrix. The results indicate that the proposed LSTM model slightly outperforms the proposed CNN model with a macro-average of precision: 0.9877, recall: 0.9876, F1 score: 0.9876. Both models demonstrate remarkable performance in accurately recognizing Pashtu numerics, achieving an accuracy level of nearly 98%. Notably, the LSTM model exhibits a marginal advantage over the CNN model in this regard.

# INTRODUCTION

Technological enhancement has catalyzed a global shift towards digitalization, where a variety of problems are addressed through computing solutions. This approach offers remarkable efficiency in tackling daily life challenges. As technology advances, it elevates computing power and intelligence, giving rise to the era of Artificial Intelligence (AI) computing. Data-driven solutions have proven to be effective in solving problems in many fields such as health sciences, cryptocurrency finance, water resources, *etc.* (*Syed*

Corresponding author
Abdulrahman Aloraini,
a.aloraini@qu.edu.sa

*et al., 2023b*; *Syed et al., 2023a*; *Syed et al., 2023c*). AI-based algorithms perform diverse tasks effectively, that are challenging for humans to perform manually. For linguistic problems, Natural Language Processing (NLP) techniques play a crucial role in proposing an efficient solution. NLP technique involves speech recognition, language modeling, image captioning, question answering, and document summarization. The application of NLP is widely used around the globe as it reduces human efforts and allows machines to understand humans either by handwritten text as an image (image captioning) or by its voice (speech recognition). For image captioning, optical character recognition (OCR) is a popular technique in which an AI model identifies a pattern for each character image and then uses it to classify the characters.

The OCR technology is able to scan characters in order to determine the shape of these characters by edge recognition and translate them into specific characters by character recognition method (*Singh et al., 2018*). In the past couple of years, the most intriguing and intricate research areas for study are in the field of image processing and pattern recognition in recognition of handwritten scripts. Many applications are relevant to this field like OCR, pattern classification, form data entry, linguistic script recognition, *etc*. Character recognition proved to be important for humans due to their speed and accuracy in recognizing scripts. For solving the problem efficiently, they rely on data-driven approaches. Among the cursive languages of South-Central Asia, Pashtu is a hugely popular language in countries such as Afghanistan, Pakistan, and Iran. In 1936, it has become a national language of Afghanistan and is being used for communication for over 35 million people most of them from Afghanistan and Pakistan while also including the smaller Pashtu communities in Iran, the United Kingdom, and United Arab Emirates (UAE). The numerals of Pashtu, Urdu, Persian, and Eastern Arabic had very similar writing patterns; However, some of the numbers in these languages are different, which is yet again another concern.

Recognition of handwritten scripts in the domain of computer science has been followed for almost half a century now. In 1959, *Singh et al. (2018)* implemented the oldest techniques for character recognition. It is known as the analysis-by-synthesis, origins by Eden in 1968. Its the basis for syntactical approaches in optical character recognition. *Gaurav Kumar, Bhatia & Indu (2013)* discussed the advancement of preprocessing techniques where input data is comprised of simple handwritten documents and deformed images. Their study concluded that applying one preprocessing method is not enough to acquire good accuracy, yet a concoction of preprocessing methods can result in acquiring reliable accuracy. *Gautam, Sharma & Hazrati (2015)* worked on Eastern Arabic numerals by optical character recognition. *Ul-Hasan et al. (2013)* employed bi-LSTM model followed by connectionist temporal classification (CTC) as an output layer was applied for recognizing printed Urdu text. *Zand, Naghsh Nilchi & Amirhassan Monadjemi (2008)* proposed a word recognition and segmentation technique for the Persian language to address the segmentation problems of the cursive nature of the Persian language. A hidden Markov model (HMM) was applied to the Pashtu language. The implementation was done Beranek, Bolt, and Newman technologies (BBN) Byblos to recognize printed data in Pashtu documents. The results concluded that the error rate for

synthetic images is 1.6%, and the error rate for scanned pages is 2.1%, while the error for faxed pages was 3.1% (*Ahmad et al., 2015*). In a study, Recurrent neural networks (RNN), and Long Short-Term Memory (LSTM) were deployed for text recognition which achieves an accuracy of 89–94 percent (*Ahmed et al., 2016*). The study aims to use data-driven modeling for classifying Pashtu numerals from 0 to 9. This study has the following research contributions.

- Identification of Pashtu digits using Convolution Neural Networks and Long Short Term Memory model due to their adaptability to handle the complexities of Pashtu numeral recognition:
  **CNN adaptation:** CNN model can automatically learn hierarchical features from Pashtu numeral (0–9) images. This adaptation allows the model to capture intricated patterns and variations present in Pashtu numeral characters which results in improving classification accuracy.
  **LSTM temporal processing:** The LSTM model is uniquely suited for sequential data processing. Hence this model can effectively capture temporal dependencies within multi-stroke numerals, aiding in accurate classification despite variations in stroke order or style.
- Proposition of classifying handwritten Pashtu digits using data-driven modeling.
- Comparative study of CNN and LSTM models performance evaluation on Optical Character Recognition.

The study is organized in the following sections. The 'Introduction' frames the introduction and necessity of the proposed study. While 'Models' represents the theoretical review of the models being implemented. 'Proposed Character Recognition Methodology' represents the methodology of the proposed study. 'Results' is comprised of results classified by the model. While 'Classification Report' discusses the performances of each model. The concluding remarks are as given in 'Conclusion'.

# MODELS

## Convolutional neural networks

A convolutional neural network (CNN) model is a type of deep learning algorithm specifically tailored for executing computer vision problems. The model is highly efficient for automatically learning hierarchical features from input data or images through a convolutional layer that detects local patterns in an image, along with a pooling layer that downsamples the data. In OCR, CNN proves highly beneficial by recognizing characters from various fonts, styles, and sizes. Their translation-invariant nature allows them to locate characters at different positions within an image, while their robustness to distortions ensures accurate text recognition even in imperfect scanned documents or images. With these capabilities, CNNs have become a fundamental technology in OCR, enabling automated text extraction and facilitating a wide range of applications that rely on visual data analysis.

A CNN is a powerful tool for proposing OCR base solutions due to its feature learning capabilities, invariance to translation, and resilience to distortions. It plays a pivotal role

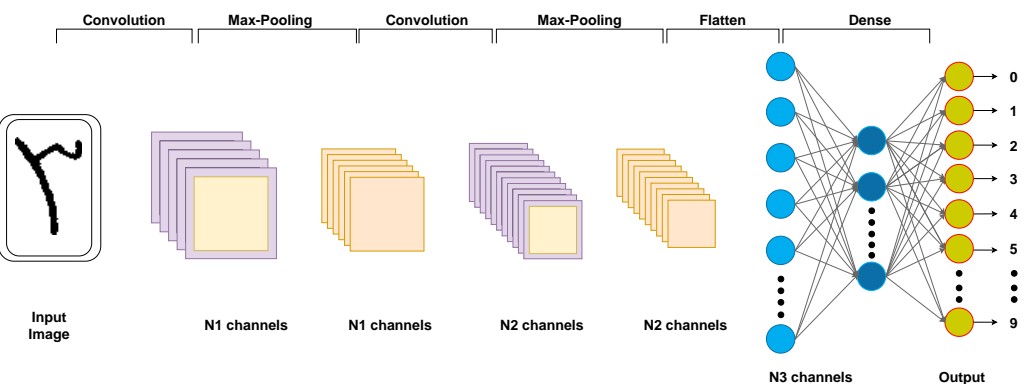

**Figure 1** **Proposed CNN model architectural scheme of the respective study.**

in automating text recognition from images, making OCR more efficient and accurate for diverse applications such as document digitization, data extraction, and content indexing. The architectural view of CNN which is illustrated in Fig. 1. The mathematical formulation for acquiring output dimensions $V_{output}$ for Convolutional layers is as shown in the following Eq. (1) (*Taye, 2023*).

$$V_{output} = 1 + \frac{V_{input} - V_{filter}}{V_{stride}} \tag{1}$$

where $V_{input}$ is the input size of an image while $V_{filter}$ and $V_{stride}$ represents the size of filter and stride.

A possible drawback of the convolution layer is the information loss at the borders of an image as $4 \times 4$ will be the output size for a $6 \times 6$ image. To overcome this shortcoming zero-padding is used. The mathematical formulation for acquiring output dimension $V_{output}$ after applying zero-padding to the image is as shown in Eq. (2) (*Taye, 2023*).

$$V_{output} = 1 + \frac{V_{input} + 2V_{pad} - V_{filter}}{V_{stride}} \tag{2}$$

where, $V_{pad}$ denotes the zero-padding layer size implied to the image.

## Long-short term memory

A long short-term memory (LSTM) is a kind of RNN model which is designed to handle sequential data by efficiently retaining and capturing long term dependencies. LSTM's ability to handle variable-length sequences, capture contextual information, and learn from raw data without extensive feature engineering makes it highly effective in OCR tasks. It excels in recognizing text from images, handling noisy data, and dealing with cursive handwriting due to its capacity to learn and remember relevant patterns across sequential information, resulting in more accurate and robust text recognition capabilities.

An LSTM model is composed of memory cells with gate components such as Control gate ($V_{control}$), Forget gate ($V_{forget}$), input gate ($V_{input}$), and output gate ($V_{output}$) whose mathematically formulations is expressed in the following equations (*Absar et al., 2022*).

$$V_{input} = \sigma(w_s \times |h_{n-1}, x_t| + b_i). \tag{3}$$

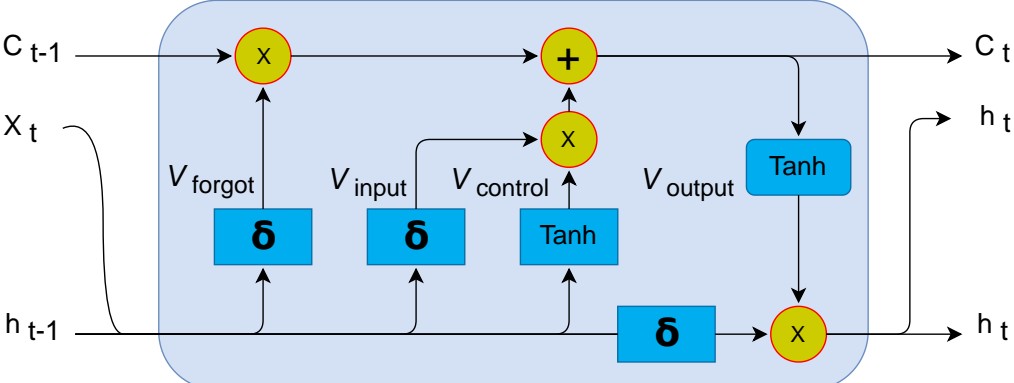

**Figure 2** Proposed LSTM unit architectural illustration for the proposed method.

The input gate is used to determines the transferability of information from a recent cell to the present cell which is mathematically explained in Eq. (3).

$$V_{forget} = \sigma(w_f \times |h_{n-1}, x_t| + b_f). \tag{4}$$

The forget gate is defined as the previous memory of the input is stored which is mathematically represented as Eq. (4).

$$V_{control} = V_{forget} \times c_{n-1} + V_{input} * c_n. \tag{5}$$

The control gate is used to modify the current cell which is mathematically explained in Eq. (5).

$$V_{output} = \sigma(w_c \times |h_{n-1}, x_n| + b_o). \tag{6}$$

The output gate is used to indicate the next hidden state as mathematically explained by Eq. (6).

$$h_n = V_{output} \times tanh(c_n) \tag{7}$$

in which $w$ is representing the weight for the matrix corresponding to the input, while $b$ is representing the bias value of a respective input.

In Fig. 2 the LSTM unit consists of three main inputs, in which $X_n$ is representing the input for the current time stamp, while $h_{n-1}$ is representing the previous LSTM's cell output. $C_{n-1}$ is denoting the previous cell memory, while $C_n$ and $h_n$ is the memory and output of the current Unit.

## PROPOSED CHARACTER RECOGNITION METHODOLOGY

The proposed method consists of a number of discrete techniques employed for Pashtu numerical character recognition. These techniques include data curation, image preprocessing, and model training, as illustrated in Fig. 3. The subsequent sub-sections provide detailed explanations of the individual techniques utilized in this proposed study.

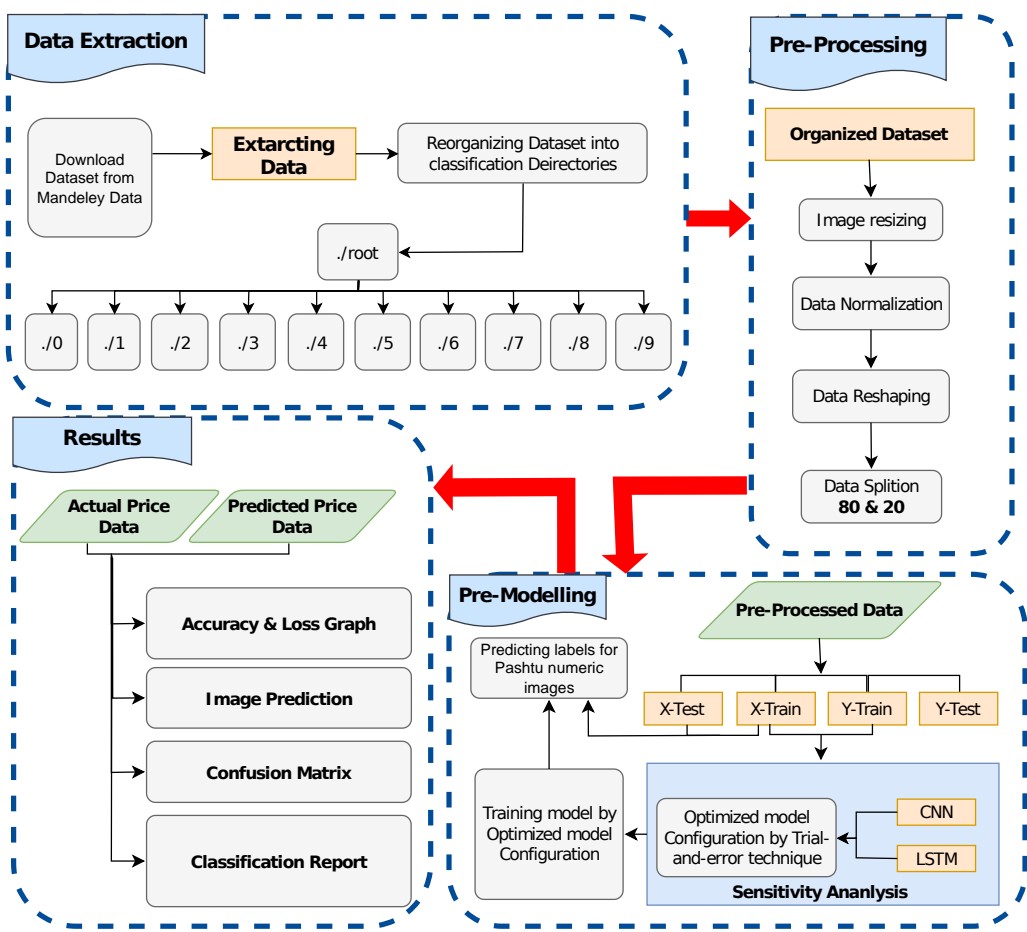

**Figure 3** Graphical illustration of proposed character recognition method.

## Data curation

A data curation is a process of carefully selecting, organizing and preparing data for use in OCR. It is crucial in OCR because high-quality curated data ensures better accuracy and reliability of the OCR system. The data used in the study is prepared by Khalil Khan and is available on Mandeley Data's open-source repository (*Khan, 2022*).

The main dataset is comprised of a total of 1,250 unique handwritten samples, each containing four sets of numerals from 0 to 9. This deliberate inclusion of multiple sets of handwriting styles aims to augment the dataset's diversity, optimizing the training environment for the model. These diverse samples are instrumental in enhancing the model's ability to generalize across different styles of Pashtu numeral handwriting. Each image captures a close-up view of the handwritten numerals, ensuring detailed and comprehensive representation in the dataset.

The data preparation for the proposed study is initiated by organizing the data in 9 directories each indicating a different Pashtu numeric for efficient classification of

numbers. The data is manually moved from the original directories to the new directories. After organizing the dataset, the data was uploaded to google drive for further analysis.

## Image processing

Image processing refers to the application of various techniques to enhance and optimize images before feeding them to an AI model. Its importance lies in improving the accuracy and efficiency of AI base OCR model by reducing noise, enhancing contrast, and standardizing image characteristics, thereby enabling better character recognition results. For Processing of images, Graphical Processing Unit (GPU) available in Google Colab was utilized due to the substantial data size. The Data was uniformized by resizing the images into $32 \times 32$ size. This step was done to ensure computational efficiency and reduce data redundancy. It simplifies feature extraction and optimizes model training and memory usage.

The data was then normalized by dividing the pixel rate of each image by 255. The was done to scale pixel values in a standardized range of [0, 1], which facilitates consistent and efficient processing of data. The mathematical representation of normalization is as shown in Eq. (8) (*Nayak, Misra & Behera, 2013*).

$$V_{new} = low + \frac{(high - low) \times (V_{old} - V_{min})}{V_{max} - V_{min}} \tag{8}$$

where $V_{new}$ is representing a new pixel value for a original pixel value $V_{old}$ in the range of [*low*, *high*], while $V_{max}$ and $V_{min}$ is representing maximum and minimum pixel value in an image.

Data reshaping is a pivotal step tailored to the unique requirements of our CNN and LSTM models. The data was also reshaped for CNN and LSTM models, separately. Reshaping for CNN and LSTM involves transforming the input data into appropriate formats suitable for each architecture. For CNN, the input data is reshaped to a 3-Dimensional tensor (height × width × channels), while for LSTM, the input data is also reshaped into a 3-Dimensional tensor (batch size × time steps × features). These adaptations are crucial to harness the specific strengths of CNN and LSTM in capturing spatial and temporal patterns, respectively, essential for accurate Pashtu numeral recognition.

Finally, the dataset was splitted into two parts, testing and training data with a ratio of 20:80, respectively. The Variables $X_{train}$, $X_{test}$ represents the training images and their labels, while variables $Y_{train}$, $Y_{test}$ represents the testing images and their labels.

## Model training

OCR requires an optimized model configuration, achieved through the trial-and-error technique, for its success. By iteratively experimenting with various configurations and hyperparameters, the OCR model's accuracy and adaptability can be significantly improved. This approach ensures the model can handle diverse input types, fonts, and Pashtu letters efficiently, resulting in better resource utilization and faster development. Moreover, analyzing errors during the process helps identify common challenges and refine the system for specific domains, making it more robust and accurate. As a result, the trial-and-error

**Table 1  Optimized CNN model summary for the proposed study.**

| Layers | Output shape | Params |
|---|---|---|
| $Conv2D_0$ | (None, 30, 30, 32) | 896 |
| $MaxPool2D_0$ | (None, 15, 15, 32) | 0 |
| $Conv2D_1$ | (None, 13, 13, 64) | 18496 |
| $MaxPool2D_1$ | (None, 6, 6, 64) | 0 |
| $Conv2D_2$ | (None, 4, 4, 128) | 73856 |
| Flatten | (None, 2048) | 0 |
| $Dense_0$ | (None, 64) | 131136 |
| $Dense_1$ | (None, 10) | 650 |
| Total params | 225,034 | |
| Trainable params | 225,034 | |
| Non-trainable params | 0 | |

technique is crucial in developing a high-performing OCR system capable of accurately classifying numerals from images.

For the CNN model, the layers used the Conv2D layer, MaxPool2D layer, Flatten layer, and Dense layer while the sequence of these layers and the filter size along with other hyperparameters for each layer were adjusted using a trial-and-error technique. The model summary is as shown in Table 1.

While for the LSTM model, the layers which were used are the timeDistributed (TD) layer, LSTM layer, and Dense layer. while the sequence of these layers and their hyperparameters were adjusted using a trial-and-error technique. The model summary is as shown in Table 2. The TimeDistributed layer is used in recurrent neural networks to process sequences or time-series data. It applies a specific layer independently to each time step in the sequence, preserving temporal dependencies and enabling the handling of varying sequence lengths. It is valuable for tasks like natural language processing and speech recognition, where order matters. The proposed timeDistributed layer was composed of a Conv2D layer, a MaxPooling2D layer and flatten layer with the same hyperparameters as in the proposed CNN model configuration.

For both model Rectified Linear Unit (ReLu) function was used as an activation function. Its non-linearity enables learning complex patterns in cursive characters or numerals. ReLU helps prevent vanishing gradient issues, improving convergence. It's computationally efficient and crucial for OCR's large data processing. Sparse activation reduces memory usage and aids generalization. ReLU function can be mathematically expressed as in Eq. (9) (*Hahnloser et al., 2000*).

$$f(x) = \begin{cases} x, & \text{for } x \geq 1 \\ 0, & \text{for } x < 1 \end{cases} \tag{9}$$

For both models, Sparse Categorical Cross entropy (SCCE) was applied as a loss function which is appropriate for multi-class classification. It enables us to perform multi-class classification efficiently while maintaining the variable's non-ordinality. Moreover, it

**Table 2 Optimized LSTM model summary for the proposed study.**

| Layers | Output shape | Params |
|---|---|---|
| $T - D_0$ | (None, 1, 30, 30, 32) | 896 |
| $T - D_1$ | (None, 1, 15, 15, 32) | 0 |
| $T - D_2$ | (None, 1, 13, 13, 64) | 18496 |
| $T - D_3$ | (None, 1, 6, 6, 64) | 0 |
| $T - D_4$ | (None, 1, 4, 4, 128) | 73856 |
| $T - D_5$ | (None, 1, 2048) | 0 |
| LSTM | (None, 64) | 540928 |
| $Dense_0$ | (None, 10) | 650 |
| Total params | 634,826 | |
| Trainable params | 634,826 | |
| Non-trainable params | 0 | |

eliminates the need to create dummy variables by directly encoding the target labels (*Chatterjee & Keprate, 2021*).

# RESULTS

## Image prediction

After training the models with their optimized configuration by training data *i.e.,* images from $X_t rain$ and labels from $Y_t rain$ each model was given unseen image data $X_t est$ for predicting their labels $Y_p red$. The predicted labels $Y_p red$ were then compared with the actual labels $Y_t est$ of testing images for evaluating the model. The evaluation was done by different statistical measures such as an accuracy and loss graph, confusion matrix, and classification report.

For the CNN model, the classification of 25 randomly selected images can be seen in Fig. 4. while for LSTM, the classification for 25 randomly selected images can be seen in Fig. 5. The images were not seen by the model during the training and thus making it new data for the model to predict which resembles a real-life case for the prediction.

## Accuracy and loss graph

The accuracy and loss graph visually illustrates a machine learning model's performance during training. It shows the model's accuracy, measuring its correctness in predictions, and the loss, quantifying the difference between predictions and actual values, over multiple training iterations (epochs). The graph is essential for assessing the model's progress, identifying overfitting or underfitting, implementing early stopping, guiding hyperparameter tuning, and understanding the training behavior. Monitoring these metrics helps ensure the model learns effectively and generalizes well to new data, enhancing its overall performance and reliability.

In Fig. 6, the accuracy and loss graph is presented for the proposed CNN model. The graph depicts that the training accuracy (Accuracy) and validation accuracy (Val_Accuracy) are initially close and increasing, suggesting the model is learning well. However, on 10th epochs the validation accuracy starts to decline, and training accuracy continues to improve

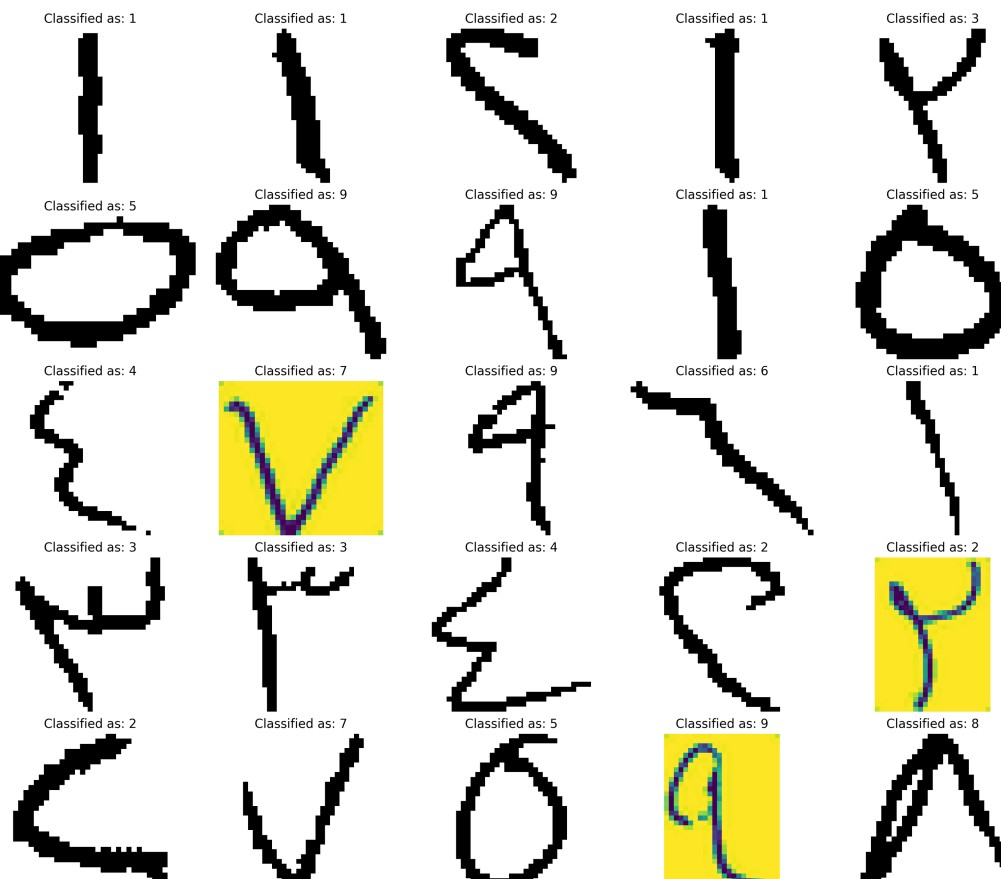

**Figure 4  CNN model numerics predictions.**

which indicates potential overfitting of the model. This divergence between training and validation accuracies suggests that while the model is effectively learning from the training data, it may struggle to generalize to unseen data. To cover these implications, regularization techniques such as dropout or weight decay could be applied, or the model architecture may require adjustments to reduce complexity. Conversely, Fig. 6 also depicts that both training and validation losses were decreased during the model's training process. However, a change occurred on the 10th epoch, where the validation loss began to increase. These observations support the decision of early stopping at epoch 10 which might be beneficial to prevent the model from becoming overly specialized to the training data and improve its generalization on unseen data.

Figure 7 illustrates the accuracy and loss graph for the proposed LSTM model. The trends in both training and validation accuracy show an initial rise, but they stabilize after the 3rd epoch, with the training accuracy showing only marginal improvement per epoch. In contrast, the loss for both training and validation data exhibits a downtrend until the third epoch, after which it starts to stabilize. Based on these observations, stopping the model training at the 10th epoch could lead to an optimized model with adequate performance.

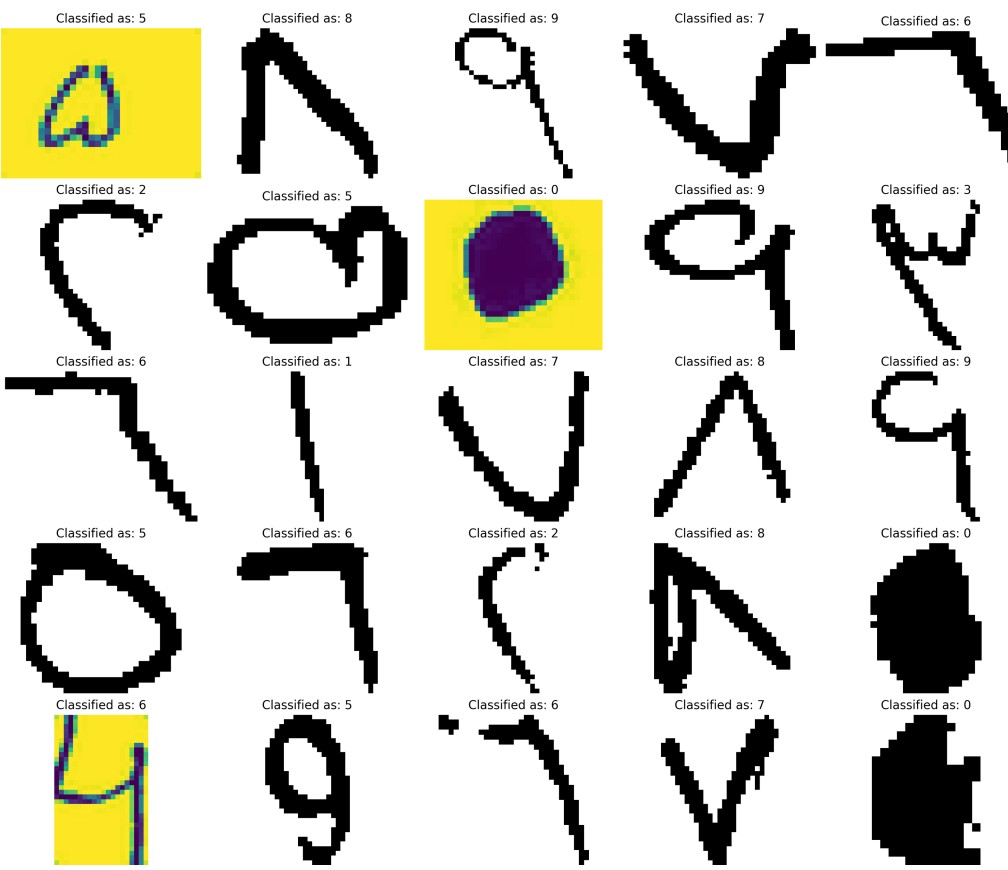

**Figure 5** LSTM model numerics predictions.

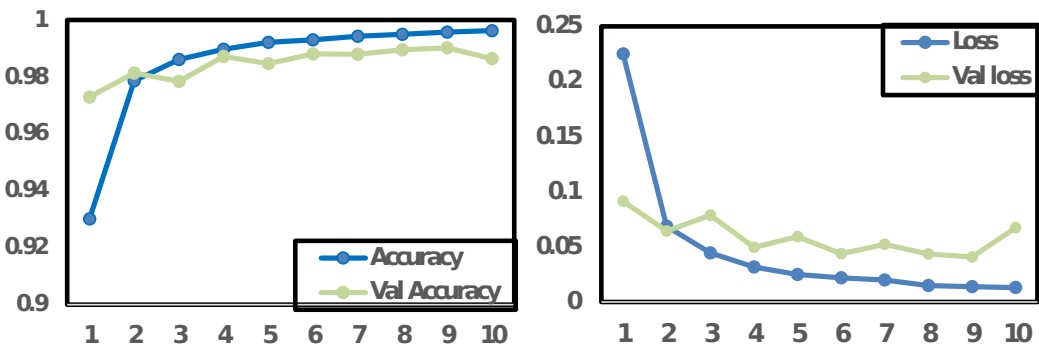

**Figure 6** Accuracy and loss graph for CNN proposed model.

## Confusion matrix

The confusion matrix is essential for evaluating the performance of a machine learning model classifying Pashtu numbers from 0 to 9. It allows us to assess accuracy, precision, recall, and identify areas of improvement. By analyzing false positives and false negatives,

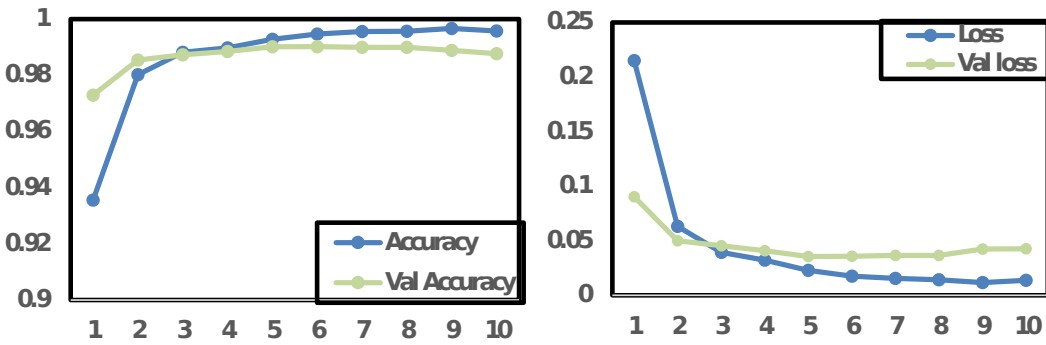

**Figure 7** Accuracy and loss graph for LSTM proposed model.

we can fine-tune the model and address class imbalances. The matrix also helps in making threshold adjustments and selecting the most suitable OCR model for optimal Pashtu number recognition.

In Fig. 8, a confusion matrix is illustrated for the CNN model predicted labels in comparison with actual labels. The results depicted in Fig. 8 demonstrate high accuracy in the model's predictions for each numeric digit. However, the area for improvement in the CNN model's performance is specifically when classifying the number '7', which is sometimes misclassified as '8'. This misclassification might be due to the visual similarity between these two digits, and it is possible that the model is sensitive to slight rotations in the input images, leading to confusion. Addressing this issue may involve exploring data augmentation techniques or fine-tuning the model to better handle such visual variations and enhance its accuracy in distinguishing between '7' and '8'.

Figure 9 illustrates the confusion matrix between the proposed LSTM model predicted labels and actual labels. As depicted in Fig. 9, the LSTM model achieved a high level of accuracy in classifying the data. Nonetheless, there is a slight need for improvement in the model's performance, specifically when recognizing the number '1', which is occasionally misclassified as '0'. This misclassification could be attributed to the visual similarity between these digits, as their primary difference lies in size and angle. Handwritten numbers can exhibit variations in angles, which may lead to misinterpretation by the model during classification. To address this issue, measures such as data augmentation, introducing rotation invariance, or fine-tuning the model to handle these visual variations more effectively can be explored to enhance the model's accuracy in distinguishing between '1' and '0'.

## Classification report

The classification report is a comprehensive evaluation of a classification model's performance metrics such as F1-score (F), recall (R), precision (P), and support (S) for all classes. It is an essential evaluator as it gives insights about a model's strengths and weaknesses for individual classes, helps in decision-making, and allows for comparisons

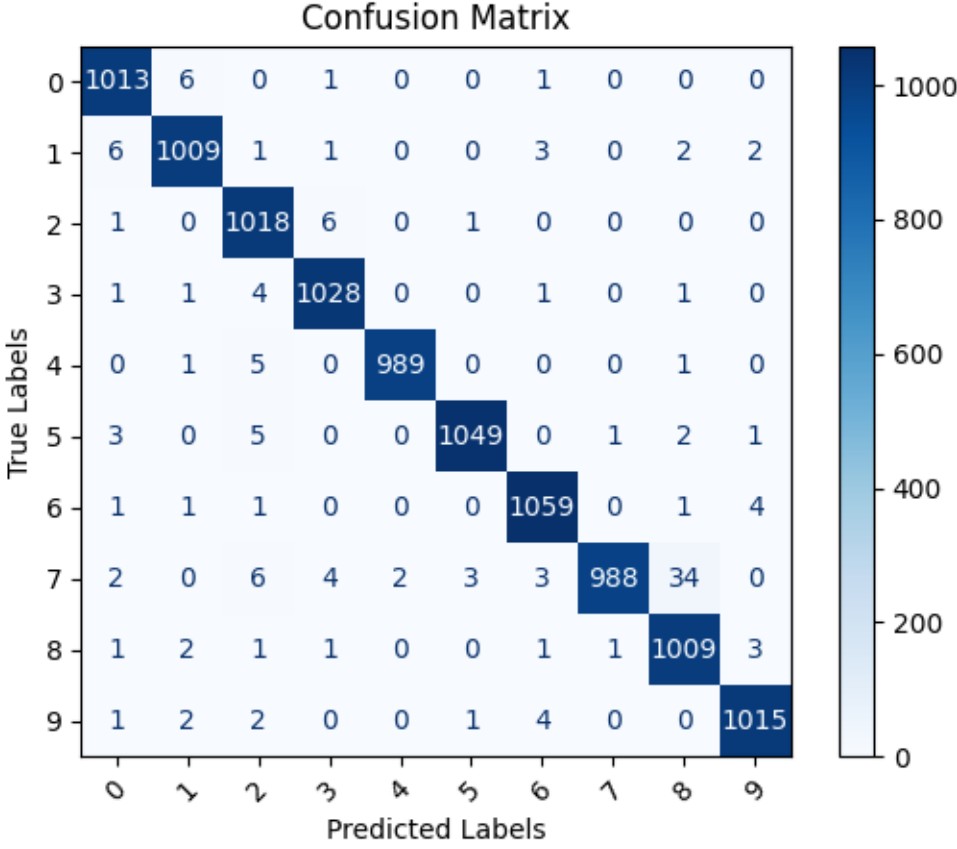

**Figure 8** Confusion matrix of actual and CNN model predicted labels.

between different models. The report assists in identifying challenging classes and guiding efforts to improve the model's accuracy and effectiveness for specific classification tasks.

Precision (P) in classification tasks is used to measure the accuracy for the positive predictions made by a model. It is calculated by measuring ratio between correctly classified samples by the model to all samples assigned to a respective class. Higher precision values signify more accurate positive predictions for the class. The mathematical representation of precision is as shown in Eq. (10) (*Hicks et al., 2022*).

$$Precision = \frac{TC}{TC + FC} \tag{10}$$

where *TC* represents classes which are classified correctly while *FC* represents falsely classified classes. The range of values can be $0 < Precision < 1$.

The recall (R) in classification tasks is used to signifies the proportion of positive samples which are correctly classified. It is calculated by measuring the ratio between correctly classified positive samples and all samples assigned to the positive class. A higher recall value indicates that the model is more adept at capturing positive instances. The mathematical representation can be seen in Eq. (11) (*Hicks et al., 2022*).

$$Recall = \frac{TP}{TP + FN} \tag{11}$$

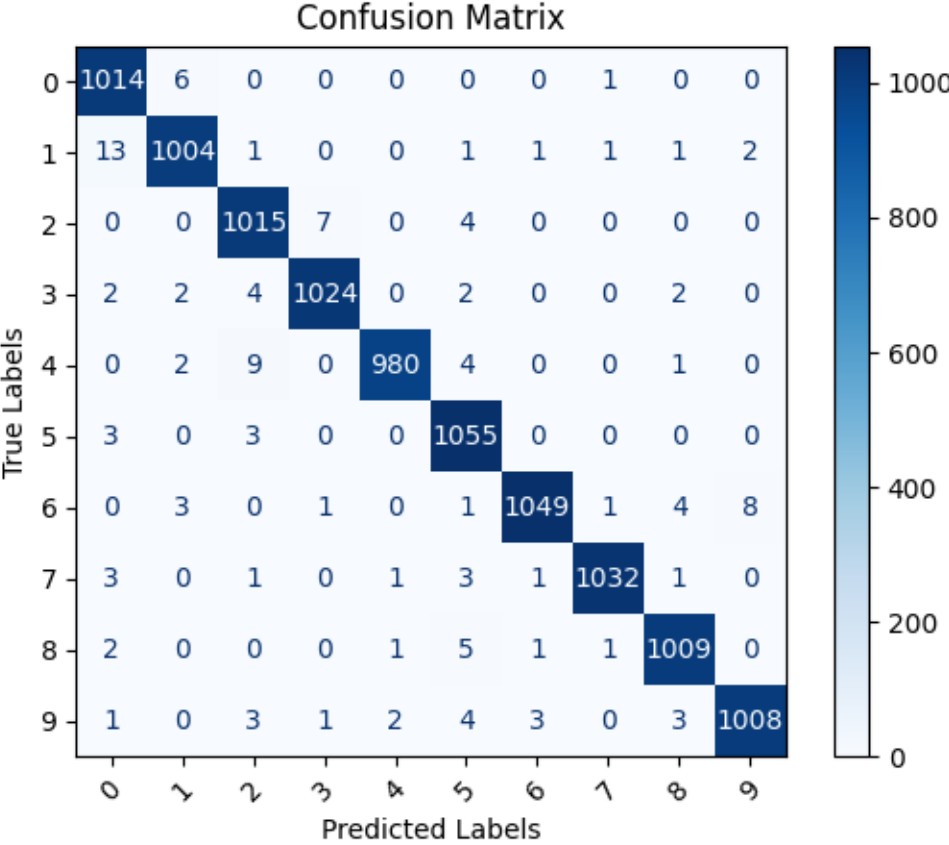

**Figure 9** Confusion matrix of actual and LSTM model predicted labels.

where *TP* is representing the number of correctly classified positive classes by the model, while *FN* is representing the number of incorrectly classified negative classes by the model. The range of values can be $0 < Recall < 1$.

The F1 score (F) is used to calculate the harmonic mean between the precision and recall of the model. By utilizing the harmonic mean, the F1 score penalizes extreme values of either precision or recall, making it an effective measure for assessing a model's overall performance. As its value varies depending on whether the class is positive or negative, F1 is not symmetric between classes. The range of values can be $0 < F1\_score < 1$. The mathematical representation is as shown in Eq. (12)?.

$$F1 - score = 2 \times \frac{Precission \times Recall}{Precission + Recall} \tag{12}$$

The support is a metric that represents the number of actual observations of the class in the specified dataset.

Accuracy is a performance evaluator which is defined as the ratio between the correctly classified samples by the model and the total number of samples in the evaluation data. In practice, however it can be misconstrued, particularly while addressing unequal class proportions. In such cases, merely assigning all samples to the prevalent class can result in

**Table 3   Pashtu numeric classification report for CNN model.**

| Numerics | Precision (P) | Recall (R) | F1-score (F) | Support (S) |
|---|---|---|---|---|
| 0 | 0.9844 | 0.9921 | 0.9882 | 1021 |
| 1 | 0.9872 | 0.9853 | 0.9863 | 1024 |
| 2 | 0.9760 | 0.9922 | 0.9840 | 1026 |
| 3 | 0.9875 | 0.9922 | 0.9898 | 1036 |
| 4 | 0.9979 | 0.9929 | 0.9954 | 996 |
| 5 | 0.9952 | 0.9886 | 0.9919 | 1061 |
| 6 | 0.9878 | 0.9925 | 0.9901 | 1067 |
| 7 | 0.9979 | 0.9481 | 0.9724 | 1042 |
| 8 | 0.9609 | 0.9901 | 0.9753 | 1019 |
| 9 | 0.9902 | 0.9902 | 0.9902 | 1025 |
| Accuracy | 0.9864 | 0.9864 | 0.9864 | 0.9864 |
| Macro avg | 0.9865 | 0.9864 | 0.9864 | 10317 |
| Weighted avg | 0.9865 | 0.9864 | 0.9864 | 10317 |

high accuracy, even though the model's performance may be inadequate. The mathematical representation is as shown in Eq. (13)?.

$$Accuracy = \frac{TP + TN}{TP + TN + FP + FN} \tag{13}$$

where *TN* represents the number of negative classes which are correctly classified, while *FP* represents the positive classes that are incorrectly classified by the model. The range of values can be $0 < Accuracy < 1$.

Macro-average (macro avg) refers to the final averaged metric that is calculated by taking the average of all classes. While the weighted-average (weighted avg) weights the contributions of each class in proportion to its size. Tables 3 and 4 represent the classification report for the proposed CNN and LSTM models, respectively. The performance metrics are computed individually for each class, enabling a comprehensive evaluation of the model's classification performance for each specific class. By comparing both models' overall performance, LSTM performs slightly better than the CNN model with a higher macro-average (Precision: 0.9877, Recall: 0.9876, F1 score: 0.9876) as compared to macro-average (Precision: 0.9865, Recall: 0.9864, F1 score: 0.9864) of CNN model.

# CONCLUSIONS

The proposed study presented an efficient way of identifying Pashtu numerics using deep learning models like CNN and LSTM. The method involves first rearranging the image data into directories representing their labels. The images were then passed on from different preprocessing activities such as image resizing, normalization, data reshaping, and data splitting. Once the data is preprocessed, it is passed on for training an optimized CNN and LSTM models. Both models perform robustly in predicting the correct labels for the images in the testing phase. The confusion matrix results showed there is slight room for improvement in both models for classifying specific digits (for CNN: 7 and for LSTM: 1).

**Table 4  Pashtu numeric classification report for LSTM model.**

| Numerics | Precision (P) | Recall (R) | F1-score (F) | Support (S) |
|---|---|---|---|---|
| 0 | 0.9768 | 0.9931 | 0.9849 | 1021 |
| 1 | 0.9872 | 0.9804 | 0.9838 | 1024 |
| 2 | 0.9797 | 0.9892 | 0.9844 | 1026 |
| 3 | 0.9912 | 0.9884 | 0.9898 | 1036 |
| 4 | 0.9959 | 0.9839 | 0.9898 | 996 |
| 5 | 0.9777 | 0.9943 | 0.9859 | 1061 |
| 6 | 0.9943 | 0.9831 | 0.9886 | 1067 |
| 7 | 0.9961 | 0.9904 | 0.9932 | 1042 |
| 8 | 0.9882 | 0.9901 | 0.9892 | 1019 |
| 9 | 0.9901 | 0.9834 | 0.9867 | 1025 |
| Accuracy | 0.9876 | 0.9876 | 0.9876 | 0.9876 |
| Macro avg | 0.9877 | 0.9876 | 0.9876 | 10317 |
| Weighted avg | 0.9877 | 0.9876 | 0.9876 | 10317 |

The classification report results showed that the LSTM model performs slightly better than the CNN model by accuracy, macro average, and weight average. The study concluded that both models are very effective in classifying Pashtu numerics.

These models could be applied to other cursive languages' numerics. The future work for the study could be to imply proposed models on different letters or numerics of the Pashtu language along with investigating other Data-driven models' behavior on the given dataset. A study could be conducted to classify real-time numerical information and subsequently transform it into textual data suitable for translation into various languages, facilitating further usage, evaluation, and analysis.

### Funding
The Prince Sultan University funded the Article Processing Charges (APC) of this publication. The funders had no role in study design, data collection and analysis, decision to publish, or preparation of the manuscript.

### Grant Disclosures
The following grant information was disclosed by the authors:
The Prince Sultan University.

### Competing Interests
The authors declare there are no competing interests.

### Author Contributions
- Sibtain Syed conceived and designed the experiments, performed the experiments, analyzed the data, performed the computation work, prepared figures and/or tables, authored or reviewed drafts of the article, and approved the final draft.

- Khalil Khan analyzed the data, authored or reviewed drafts of the article, and approved the final draft.
- Maqbool Khan conceived and designed the experiments, authored or reviewed drafts of the article, and approved the final draft.
- Rehan Ullah Khan analyzed the data, authored or reviewed drafts of the article, and approved the final draft.
- Abdulrahman Aloraini conceived and designed the experiments, authored or reviewed drafts of the article, and approved the final draft.

## Data Availability

The raw data is available at Mendeley data repository: Syed, Sibtain; Khan, Khalil (2024), "Pashtu Numerals (0-9)", Mendeley Data, V1, doi: 10.17632/27ywyyc54f.1.

The code is available in the Supplemental File.

## Supplemental Information

Supplemental information for this article can be found online at http://dx.doi.org/10.7717/peerj-cs.2124#supplemental-information.

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
