# Peer review of "Recognition of inscribed cursive Pashtu numeral through optimized deep learning"

_PeerJ Computer Science, doi:10.7717/peerj-cs.2124_

## Round 0.1 · original submission · Major Revisions

Clear the contributions of work. Address all the comments of reviewers.
**Language Note:** The review process has identified that the English language must be improved. PeerJ can provide language editing services - please contact us at [email protected] for pricing (be sure to provide your manuscript number and title). Alternatively, you should make your own arrangements to improve the language quality and provide details in your response letter. – PeerJ Staff

Reviewer 1 ·

Basic reporting

The paper titled "Recognition of Inscribed Cursive Pashtu Numeral Through Optimized Deep Learning" endeavors to introduce a deep learning methodology for the identification of Pashtu numerals ranging from 0 to 9. While the paper appears promising, yet certain concerns have also been raised for the authors to address.
• Mention the characteristics of the dataset, such as the diversity of handwriting styles, or any data quality issues, etc. for assessing the generalizability and reliability of the proposed model.
• A brief rational explanation for the necessity of data reshaping is required to enhance clarity for readers unfamiliar with the technical aspects of these architectures.
• The introduction uses terms like "Pashtu" and "Pashto" interchangeably. Ensuring consistency in terminology will contribute to a clearer and more professional presentation of the research.
• The article should remove the grammatical mistakes. Please proofread paper for small grammatical mistakes and such as using capital letter in the middle of the sentences.
• The article requires overall improvements in English language usage and flow.
• I would recommend acceptance upon addressing these issues and enhancing the clarity and coherence of the article.

Experimental design

• Mention the characteristics of the dataset, such as the diversity of handwriting styles, or any data quality issues, etc. for assessing the generalizability and reliability of the proposed model.
• A brief rational explanation for the necessity of data reshaping is required to enhance clarity for readers unfamiliar with the technical aspects of these architectures.

Validity of the findings

• The results and discussion section lacks analysis and details. Add implication of your results too.

Additional comments

• The introduction uses terms like "Pashtu" and "Pashto" interchangeably. Ensuring consistency in terminology will contribute to a clearer and more professional presentation of the research.
• Formatting of Table 1 and Table 2 is different. Please use the right formatting according to the journal.

Cite this review as

Reviewer 3 ·

Basic reporting

This paper proposes a CNN and LSTM based model to classify pashtu handwritten words, This problem looks important since there has been very less work pashto language. However, the paper need much more improvements;

I don't see the contribution of the paper is comprehensively described? It is better to explain in more detail. such as you say CNN and LSTM models are used. However, this not a contribution, the point is what modification you made to lstm and CNN for your application. need to explain in a little more detail.

too many typos and awkward mistake in throughout the manuscript such as after eq. 1 .....where Vinput... what is Vinput?
what is the question mark before equation 3 .... after applying zero-padding to the image
is as shown in equation 2 ?..... the title of subsection image Processing? ...........the question mark before equation 11......

it sounds like the authors created their own dataset right? if i am not missing something this could be considered as your contribution.

the recall and precision formulas are quite strange. why don't you use TP,TNFPFN for both formulas

Experimental design

nn

Validity of the findings

nn

Additional comments

nn

Cite this review as

---

## Round 0.2 · accepted · Accept

Paper has been improved and now acceptable for publication.

Reviewer 1 ·

Basic reporting

All of my concerns are addressed adequately, so I do not have any further comments. I would recommend acceptance of the article in the current state.

Experimental design

well defined

Validity of the findings

Clearly stated and executed

Additional comments

NA

Cite this review as